# The Value of C-Reactive Protein and Peritoneal Cytokines as Early Predictors of Anastomotic Leak after Colorectal Surgery

**DOI:** 10.3390/diagnostics14182101

**Published:** 2024-09-23

**Authors:** Dubravka Mužina, Mario Kopljar, Zdenko Bilić, Blaženka Ladika Davidović, Goran Glavčić, Suzana Janković, Monika Mačkić

**Affiliations:** 1Department of Surgery, University Hospital Center Sisters of Charity, 10000 Zagreb, Croatia; zbilic84@gmail.com (Z.B.); glavcic.goran@gmail.com (G.G.); suzana.jankovic33@gmail.com (S.J.); monika.mackic@gmail.com (M.M.); 2Faculty of Medicine Osijek, University Hospital Sisters of Charity, Zagreb and Josip Juraj Strossmayer University of Osijek, 10000 Zagreb, Croatia; kopljar@yahoo.com; 3Department of Oncology and Nuclear Medicine, University Hospital Center Sisters of Charity, 10000 Zagreb, Croatia; blazenka.ladika.davidovic@kbcsm.hr

**Keywords:** colorectal carcinoma, colorectal surgery, anastomotic leakage, C-reactive protein, interleukin-6, TNF-alpha

## Abstract

Objectives: The aim of this study was to evaluate the accuracy of serum C-reactive protein (CRP) and intraperitoneal CRP, interleukin-6, and tumor necrosis factor-alpha in early diagnostics of anastomotic leakage in the first 4 postoperative days after colorectal surgery. Methods: Between January 2023 and June 2023, one hundred patients with colorectal carcinoma were operated on with primary anastomosis. Ten patients had anastomotic leak (10%). Results: Based on serum CRP, a patient with a leak will be detected with a 78% probability on postoperative day 3 with values above 169.0 mg/L and on postoperative day 4 with values equal to 159.0 mg/L and above. Intraperitoneal CRP values greater than 56 mg/L on the fourth postoperative day indicate a 78% probability of a diagnosis of leakage. An anastomotic leak will be detected with a 70.0% probability based on an IL-6 value on the first day, at a cut-off value of 42,150. The accuracy of TNF-alpha in predicting anastomotic leak in the first two days is 70% at values higher than 78.00 on the first and 58.50 on the second postoperative day. Conclusion: In this study serum CRP proved to be the most accurate in predicting anastomotic dehiscence after colorectal surgery.

## 1. Introduction

The optimal treatment of colorectal cancer is surgical resection and primary anastomosis. The frequency of anastomotic leak varies from 2 to 26%, which mostly depends on the location of the anastomosis [1,2,3,4]. Ileocolic anastomosis has the lowest percentage of anastomotic leak, from 1 to 3% [5,6]. The highest percentage of anastomotic leak is detected in patients with low coloanal anastomosis, ranging from 10–26% [7]. Overall, the majority of dehiscences (75%) occur on colorectal anastomoses [8]. Most anastomotic leaks, according to several large studies, are diagnosed between the 5th and 8th postoperative days [8,9,10].

Failure of intestinal anastomosis is associated with higher mortality and morbidity rates, a higher number of repeated surgical procedures, more radiological interventions, a higher percentage of permanent stoma, increased hospital stay, higher treatment costs, a higher percentage of local disease recurrence, and shorter overall survival in oncological patients [4,5,6,7,11]. Anastomosis leak is the leading cause of death after colon surgery, with a mortality rate of 6–22% [12,13,14] due to septic complications. In the first days after surgery, clinical signs and symptoms of anastomotic leak (vague abdominal pain, intestinal paralysis, fever, elevated serum CRP levels) are difficult to distinguish from the normal systemic inflammatory response to surgery. Delayed diagnosis and postponement of adequate intervention significantly increase patient mortality. A 2.5-day delay in diagnosing anastomotic leakage in a study by van Dulk et al. increased patient mortality from 24% to 39% [15]. Given the unreliability of clinical signs, surgeons often use inflammatory parameters (leukocytes and serum CRP) to evaluate the postoperative course after colorectal surgery. Unfortunately, these markers are also non-specific since their values rise in secondary infections that often occur postoperatively (e.g., urinary tract infection, respiratory tract infection, surgical site infection). With clinical suspicion of anastomotic leakage, the most commonly used radiological method of diagnosis is abdominal MSCT. The problem is that the MSCT of the abdomen has relatively low sensitivity and specificity in early diagnosis of anastomotic leak after colorectal surgery. One of the studies that investigated the accuracy of abdominal and pelvic MSCT in the diagnosis of dehiscence showed an 84% accuracy for anastomotic leak. The same research showed that a false negative MSCT finding leads to a significant delay in the treatment of dehiscence and a tenfold increase in mortality—from 4.2% to 45.5% mortality with a delayed diagnosis of anastomotic leak after a false-negative MSCT scan [16].

Early diagnosis of anastomotic leak after colorectal surgery can reduce the adverse effects of AL by reducing morbidity and mortality. There are still no widely accepted biomarkers or scoring system that can clearly identify patients at risk for anastomotic leak after colorectal surgery. Currently, diagnosis of an anastomotic leak is based on clinical suspicion and a subsequent abdominal MSCT scan. This study was designed to assess the practical utility of serum and peritoneal C-reactive protein (CRP) and peritoneal cytokines, interleukin-6 (IL-6), and tumor necrosis factor-alpha (TNF-alpha), in early diagnosis of anastomotic leak, before evident clinical signs appear.

## 2. Material and Methods

### 2.1. Patients

Between January 2023 and June 2023, all patients undergoing an elective operative treatment for colorectal cancer within the oncologic surgery department of the University Hospital Sisters of Charity in Zagreb, Croatia, were taken into account for participation in this study. Inclusion criteria were pathologically confirmed diagnosis of colorectal cancer, planned primary anastomosis, age ≥ 18 years, and capacity to consent. Exclusion criteria were age younger than 18 years, neoadjuvant chemoradiotherapy for colorectal cancer, patients operated in emergency settings, incapacity to consent, and a formation of the colostomy. All surgical procedures were performed by the same surgical team. Surgical procedures were performed either laparoscopically or through a midline laparotomy, depending on the surgeon’s preference. Stapling devices from the same manufacturer were used to form the anastomosis. At the end of the procedure, an intra-abdominal non-suction drain was placed near the anastomosis and was removed when the last sample was taken for the peritoneal fluid analysis on the 4th postoperative day. Excluding the placement of intra-abdominal drains, the perioperative course was carried out in all patients according to the ERAS protocol. In this study, anastomotic leakage was defined by clinical and radiological signs: purulent or fecal discharge from the abdominal drain, clinical signs of peritonitis or sepsis, presence of air or abscess near anastomosis on MSCT scan of the abdomen and pelvis, and presence of fecal or purulent content near anastomosis during reoperation.

### 2.2. Sample Collection

Blood samples and intraperitoneal fluid were collected at days 1–4 postoperatively at the same time of the day. For the evaluation of CRP levels in serum and peritoneal fluid, 6 mL of blood and 9 mL of intraperitoneal fluid were collected. Serum and intraperitoneal levels of CRP were measured by latex immunoturbimetry, in a routine manner, in the laboratory in our hospital. A sample of intraperitoneal fluid (9 mL) for the determination of TNF-alpha and IL-6 levels was analyzed in the laboratory of the Clinic of Oncology and Nuclear Medicine in University Hospital Sisters of Charity. Intraperitoneal fluid samples were centrifuged at 3000× *g* for 15 min at 4 °C immediately after collection. The supernatant was stored at −80 °C until analysis. The values of IL-6 and TNF-alpha were measured with an immunometric chemiluminescent assay (Immulite 1000, Siemens, Munich, Germany). The calibration of IL-6 was from 4 to 1000 pg/mL, and from TNF-alpha from 2 to 1000 pg/mL.

### 2.3. Statistics

The data were analyzed using the statistical package SPSS software (IBM SPSS Statistics for Windows, Version 24.0. Armonk, NY, USA: IBM Corp.). Descriptive statistics were used to describe the sample.

Categorical variables were analyzed by chi-square test (with the Fisher exact significance level for 2 × 2 contingency tables), and Mann–Whitney nonparametric tests for independent samples were used to check the significance of differences between two groups of patients in measured parameters (continuous variables) whose distributions deviated from the normal distribution.

In the case of repeated measurements on the same subjects, two-way and three-way ANOVA with repeated measurements were used, with the Scheffe post hoc test and Bonferroni correction for multiple comparisons.

To determine which factors are predictors of dehiscence, a series of binary logistic regressions were performed. The forward conditional method of logistic regression was used, where the criterion for including a variable in the model in each step was 0.05 and the criterion for exclusion was 0.10. Also, in order to determine what the limit value is, i.e., which is the so-called A series of receiver operating characteristic curves (ROC) was made to predict the cut-off value of individual cytokines and CRP, according to which we could predict with the highest accuracy whether the patient will have dehiscence or not (in a situation where we also know the objective state of whether he has dehiscence or not). A *p*-value of <0.05 was considered statistically significant.

## 3. Results

### 3.1. Patient Characteristics

A total of 100 patients with colorectal cancer undergoing elective rectal surgery were enrolled in this study.

The study included 56 (56.0%) male and 44 (44.0%) female patients. The mean age of the participants was 68.0 ± 14.6 years. Characteristics of patients are shown in Table 1. Ten (10%) patients developed AL. AL was diagnosed between postoperative days 2 and 10. Nine patients with anastomotic leak required repeated surgery consisting of end colostomy and abdominal drainage. In one patient, anastomotic leak was successfully treated conservatively with the use of antibiotic therapy, cessation of oral nutrition, and prolonged abdominal drainage. In a group of patients with right hemicolectomy (19 patients), no anastomotic leak occurred. In 38 patients, left hemicolectomy with colorectal anastomosis was performed, of which three patients had anastomosis leak (7.9%). Of the 32 patients operated on for rectal cancer, seven had anastomosis leak (21.9%).

One patient (1%) died in the 30-day postoperative period, in whom anastomosis leak occurred, and the cause of death was septic shock and multiorgan failure.

### 3.2. CRP Levels in Serum

It was found that the change in serum CRP over time is statistically significant (F_(1.7,166.6)_ = 8.614, *p* = 0.001). Scheffe’s post hoc test shows that there is an increase in the second and third days compared to the first day (both *p* < 0.001), without further increase.

The statistically significant effect of anastomotic leak is (F_(1.96)_ = 22.713, *p* < 0.001), indicating that the group with anastomotic leak has higher average serum CRP values compared to the group without leak. Also significant is the interaction of anastomotic leak with the change over time (F_(1.7,166.6)_ = 16.804, *p* < 0.001), which is manifested in the fact that in the group with anastomotic leak, the average values of serum CRP increase on the second and third days with a slight increase on the fourth day, while in the group without leak, serum CRP levels begin to decline on the third day and continue to decrease on the fourth day (Figure 1).

Statistically significant values are marked, where it is evident that serum CRP values are important in the detection of anastomotic leak on the second, third, and fourth postoperative days, with the highest accuracy in predicting dehiscence having levels of serum CRP on the third and fourth days (both AUC > 0.800). An anastomotic leak will be detected with a 78% probability based on serum CRP values on day 3 with values above 169.0 mg/L and on day 4 with values of 159.0 mg/L and above (Figure 2, Table 2).

### 3.3. CRP Levels in Peritoneal Fluid

It was found that the change in intrapertoneal CRP over time is statistically significant (F_(1.9,181)_ = 60.332, *p* = 0.001), and Scheffe’s post hoc test shows that there is a continuous increase every subsequent day (all *p* < 0.001).

The statistically significant effect of anastomotic leak is (F_(1.96)_ = 5.901, *p* = 0.017), indicating that the group with anastomotic leak has higher average values of intraperitoneal CRP compared to the group without leak. Also significant is the interaction of anastomotic leak with the change in intraperitoneal CRP over time (F_(1.9,181)_ = 17,830, *p* < 0.001), which is manifested in the fact that in the group with anastomotic leak, the average values of intraperitoneal CRP increase sharply each subsequent day, while in the group without leak, values of intraperitoneal CRP, after a slight increase in the second and third days, reach a plateau on the fourth postoperative day (Figure 3).

A statistically significant value was marked from which it is evident that intraperitoneal values of CRP are important in the detection of anastomotic leak only on the fourth postoperative day, with the accuracy in predicting anastomotic leak being 78% at values of 56.65 mg/L and above (AUC = 0.815, CI: 0.697–0.933) (Figure 4, Table 3).

### 3.4. IL-6 Levels in Peritoneal Fluid

It was found that the change in IL-6 over time is statistically significant (F_(1.3,125.9)_ = 5.296, *p* = 0.015), with Scheffe’s post hoc test showing that there is a significant decrease in the value of the fourth day compared to the first day (*p* = 0.047).

The effect of anastomotic leak is statistically significant (F_(1.96)_ = 3.894, *p* = 0.051), indicating that the group with anastomotic leak may have higher average values of IL-6 compared to the group without leak, but it is desirable to verify this finding in future studies on a larger sample.

There is no significant interaction of anastomotic leak with the change in IL-6 values over time (F_(1.3,125.9)_ = 0.374, *p* = 0.599), which means that the decrease in mean values occurs in a similar way in both groups of patients (Figure 5).

Statistically significant values of IL-6 in the detection of anastomotic leak on the first day (AUC = 0.782, CI: 0.661–0.893) were indicated. An anastomotic leak will be detected with a 70.0% probability based on an IL-6 value on the first day, at a cut-off value of 42,150 (Figure 6, Table 4).

### 3.5. TNF-Alpha in Peritoneal Fluid

It was found that the change in TNF-alpha over time was statistically significant (F_(1.4,136.3)_ = 3.623, *p* = 0.045), although Scheffe’s post hoc test showed that there was a significant decrease only on the third day compared to the second day.

The statistically significant effect of anastomotic leak is (F_(1.97)_ = 53.266, *p* < 0.001), indicating that the group with anastomotic leak has significantly higher average TNF-alpha values compared to the group without leak.

The interaction of anastomotic leak with the change in TNF-alpha over time (F_(1.4,136.3)_ = 3.662, *p* = 0.043) is also significant, which is manifested in the fact that in the group with anastomotic leak, the average TNF-alpha values drop sharply on the third day, while in the group without leak, TNF-alpha is relatively stable throughout all four postoperative days (Figure 7).

Statistically significant values are marked, where it is evident that the TNF-alpha value is good for detecting anastomotic leak on the first two days as well as on the fourth day of measurement (all three AUC > 0.700), with the accuracy in predicting anastomotic leak in the first two days being 70% at values higher than 78.00 on the first and 58.50 on the second day. On the fourth day, the sensitivity is slightly lower and is 56% for patients with values greater than 48.50 (Figure 8, Table 5).

## 4. Discussion

Cytokines are a group of low-molecular-weight proteins that include interleukins and interferons. They are produced at the site of injury from activated leukocytes, fibroblasts, and endothelial cells and play a significant role in the inflammatory response to injury caused by surgery. Their effect is local, where they maintain an inflammatory response at the site of tissue injury; and systemic, stimulating the formation of inflammatory phase proteins in the liver.

The early inflammatory response after surgical injury is characterized by the secretion of cytokines TNF-alpha and interleukin-1 (IL-1) from activated neutrophils and macrophages. TNF-alpha induces the secretion of IL-6, the main cytokine responsible for initiating systemic changes, that is, the acute phase response. The level of IL-6 production reflects the degree of tissue damage, and the highest values of IL-6 were recorded after major abdominal, vascular, and orthopedic surgical procedures. The maximum production of IL-6 reaches 24 h after surgery and remains elevated for the next 48–72 h. One of the main features of the acute phase response is the formation of acute phase proteins in the liver, which act as inflammatory mediators, antiproteinases, and participate in tissue repair. In acute phase proteins These include CRP, fibrinogen, alpha2-macroglobulin, and other antiproteinases. On the other hand, the formation of other proteins in the liver, such as albumin and transferrin, decreases. The physiological role of CRP is to bind to lysophosphatidylcholine, which is expressed on the surface of dead or dying cells and certain types of bacteria, thereby activating the complement system, with consequent phagocytosis by macrophages. In healthy adults, normal CRP values are less than 5.0 mg/L. After the stimulus, serum CRP values begin to rise rapidly (6 h after surgery) and reach their maximum after 48 h. The half-life of CRP in plasma is 19 h, and without stimulus, its serum values begin to drop very quickly. In addition to the formation of acute phase proteins, a feature of the acute phase response is the increased production of antibodies and effector T cells.

Healing of the anastomosis is a dynamic process that goes through three stages until complete healing. The anastomosis is weakest 48 h after formation, when collagenase activity is highest and the strength of the anastomosis gradually strengthens as new collagen is formed from fibroblasts and smooth muscle cells during the remodeling phase.

In order to reduce morbidity and mortality in anastomotic leak, it is necessary to start treatment at the earliest possible stage, in the phase of local inflammation, when clinical signs of leak are not yet apparent [17,18].

One of the most researched markers of early diagnosis of anastomotic leak is serum CRP. A meta-analysis by Singh et al., which included 2483 patients, showed that CRP is a useful negative predictor for the development of dehiscence from the 3rd to the 5th postoperative day, with a negative predictive value of 97% [19]. Another review article, by Daams et al., which included 70 studies, showed that CRP has a reasonable predictive value for anastomosis leak, with cut-off values of 120–190 mg/L measured on days 3–4 postoperatively. In the same review article, tests performed on intraperitoneal fluid from the abdominal drain were also analyzed. While the results for the predictive value of intraperitoneal CRP were inconclusive, the cytokines IL-1, IL-10, IL-6, and TNF-alpha showed that cytokine values were elevated from the first postoperative day [20]. These cytokines secrete macrophages and neutrophils that infiltrate the anastomosis area during the first postoperative day. If the healing of the anastomosis is correct, without extravasation of bacteria from the lumen of the intestine, cytokine values will quickly return to normal values. Similar results were presented in a review article and a meta-analysis by Sparreboom et al., which showed that intraperitoneal values of cytokines IL-1, IL-10, IL-6, and TNF-alpha are higher in patients with dehiscence during the early postoperative period. Particularly noteworthy is IL-6, which is significantly elevated on the first postoperative day, as well as TNF-alpha [21].

In our study, serum CRP was shown to be a reliable marker for the earlier diagnosis of anastomotic leak. In patients with anastomotic leak, serum CRP values increase continuously until the fourth postoperative day, while in patients without leak, serum CRP values rise on the second postoperative day and then begin to decrease. The highest accuracy in predicting anastomosis leak is the value of serum CRP on the third and fourth postoperative days (both AUC > 0.800). A patient with anastomotic leak will be detected with a 78% probability based on serum CRP values on postoperative day 3 with values above 169.0 mg/L and on postoperative day 4 with values equal to 159.0 mg/L and above. In accordance with other studies [19,20], this study has shown that repeated measurement of serum CRP in the early postoperative period is a valuable tool in the timely diagnosis of anastomotic leak and can give surgeons insight into the healing of intestinal anastomosis.

In patients with anastomotic leak, a continuous increase in CRP values in the intraperitoneal fluid was monitored, while in patients without leak, the values increase on the 2nd and 3rd postoperative days and then reach a plateau on the 4th postoperative day. In the prediction of anastomotic leak, the values of only the fourth day are important, with the accuracy in the prediction of anastomotic leak being 78% at values of 56.65 mg/L and above. Although the accuracy of the measured values is high enough to be a good marker for early diagnosis, the disadvantage of intraperitoneal CRP is a “delay” for serum CRP values by one day. Given that CRP is produced in the liver and intraperitoneal fluid is a plasma ultrafiltrate, there is a physiological explanation for this 24 h delay in values of intraperitoneal CRP.

Previous studies have not provided a conclusive conclusion on the value of IL-6 and TNF-alpha in the diagnosis of anastomotic leakage. A meta-analysis involving eight studies showed that intraperitoneal values of IL-6 and TNF-alpha were statistically significantly higher in the first two postoperative days in patients who will have anastomosis dehiscence [22]. On the other hand, a meta-analysis involving 16 studies did not show that peritoneal IL-6 is a valuable marker in the early diagnosis of anastomotic leak [23]. One randomized study by Mari et al. showed that implementing ERAS protocols in patients decreases serum IL-6 and CRP values after laproscopic surgery for colorectal cancer [24].

In our study, the values of IL-6 from the intraperitoneal fluid show an increase in the first postoperative day in both groups of patients, with the values being higher in patients with anastomotic leak. In both groups, the decrease in values is monitored until the 4th postoperative day. In the prediction of anastomotic leak, the measurement of the level of IL-6 on the first postoperative day has the greatest value. A patient with anastomotic leak will be detected with a 70.0% probability based on the IL-6 value on the first day, at a cut-off value of 42,150. In addition, in this study, six patients who had IL-6 values greater than 100,000 on the first postoperative day developed after 2–6 days clear clinical signs of anastomotic leak.

Measurement of TNF-alpha values in patients with anastomotic leak showed an increase in the first two postoperative days, followed by a decrease in TNF-alpha values in perioneal fluid. In patients without leak, TNF-alpha values were very similar on all four days of measurement. For the prediction of anastomotic leak, accuracy of TNF-alpha in the first two postoperative days is 70% at values higher than 78.00 on the first and 58.50 on the second postoperative day.

There are few limitations to this study. First, although serum CRP is the most widely used biomarker, defined cut-off values for anastomotic leak diagnosis do not exist. Given that any inflammatory conditions in the body (infections, neoplasms, autoimmune conditions, surgery, trauma, burns, et cetera) raise the value of serum CRP, there is a possibility of false-positive results, so it is necessary to exclude some other, unrelated inflammatory process before a definitive diagnosis of anastomotic leak. On the other hand, some patients do not have the ability to produce CRP, especially in immunocompromised cases, leading to false-negative results [25].

The limitation considering IL-6 and TNF-alpha as biomarkers for early diagnosis of anastomotic leak, according to previous research, is that there are no generally accepted basic levels for these cytokines, as there are for CRP, which makes it difficult to correctly interpret changes in IL-6 and TNF-alpha levels in peritoneal fluid during the early postoperative course.

## 5. Conclusions

This study showed that measuring serum CRP and intraperitoneal CRP, IL-6, and TNF-alpha during the first four postoperative days can give surgeons valuable and reliable insight into the healing of intestinal anastomosis. It was particularly interesting to find that extremely high values (above 100,000) of intraperitoneal IL-6 on the first postoperative day indicated anastomotic leak in the coming days. Serum CRP proved to be the most accurate in predicting anastomotic dehiscence. In smaller hospitals that do not have cytokine measurement equipment, serum CRP is a reliable, inexpensive, and widely available marker for the early diagnosis of anastomosis leak, especially on the 3rd and 4th postoperative days.

The limitation of this study is that this was a single-center study with a small sample size; therefore, prospective multicenter studies with a greater number of patients are necessary considering the lack of cut-off values for serum CRP and the lack of basic and cut-off values for IL-6 and TNF-alpha in peritoneal fluid. Extended research is necessary to further evaluate our findings and possibly extend them to daily clinical practice.

## Figures and Tables

**Figure 1 diagnostics-14-02101-f001:**
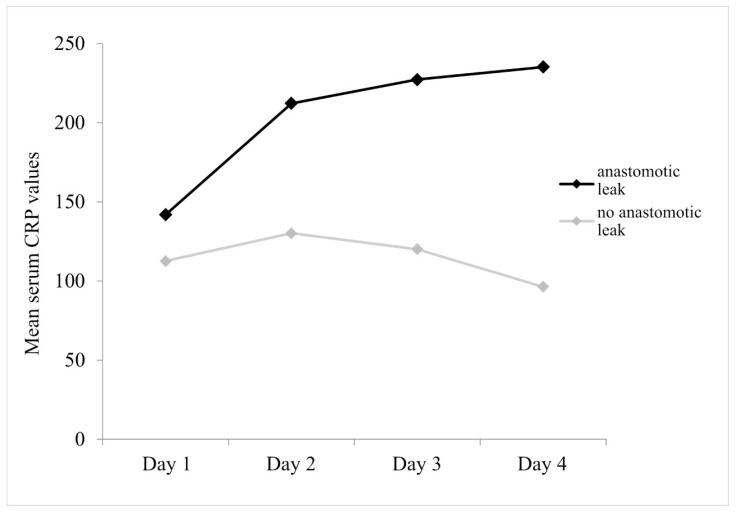
Mean serum CRP values over 4 days of follow-up, with respect to anastomotic leak.

**Figure 2 diagnostics-14-02101-f002:**
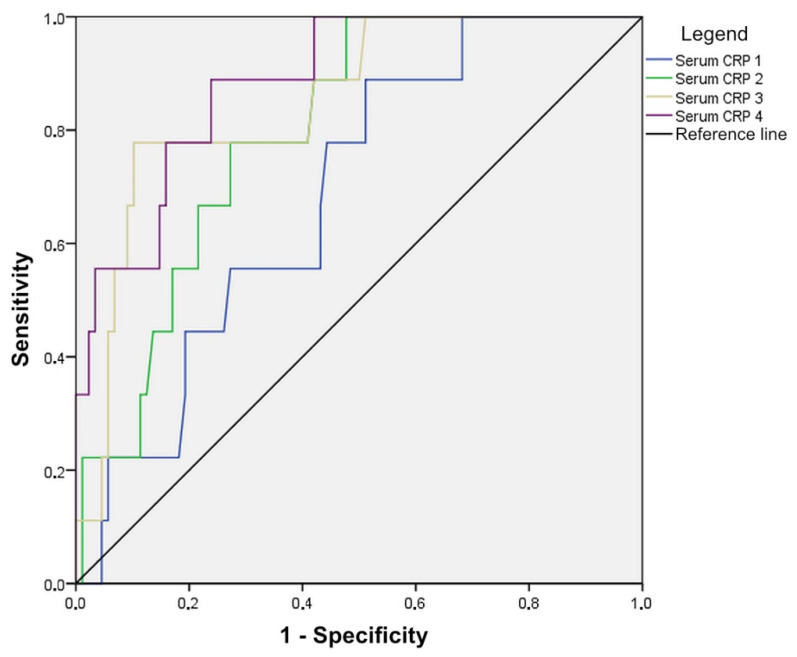
Diagnostic accuracy of serum CRP in detecting anastomotic leak, shown through ROC curves.

**Figure 3 diagnostics-14-02101-f003:**
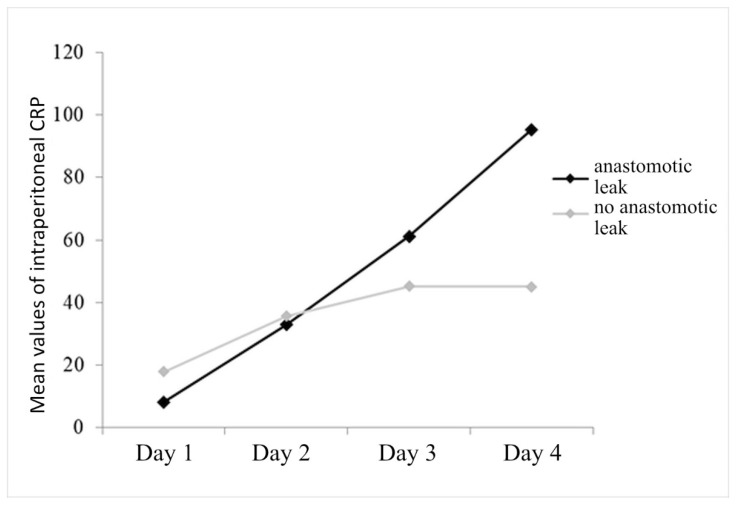
Mean values of intraperitoneal CRP at 4 days of follow-up, with respect to anastomotic leak.

**Figure 4 diagnostics-14-02101-f004:**
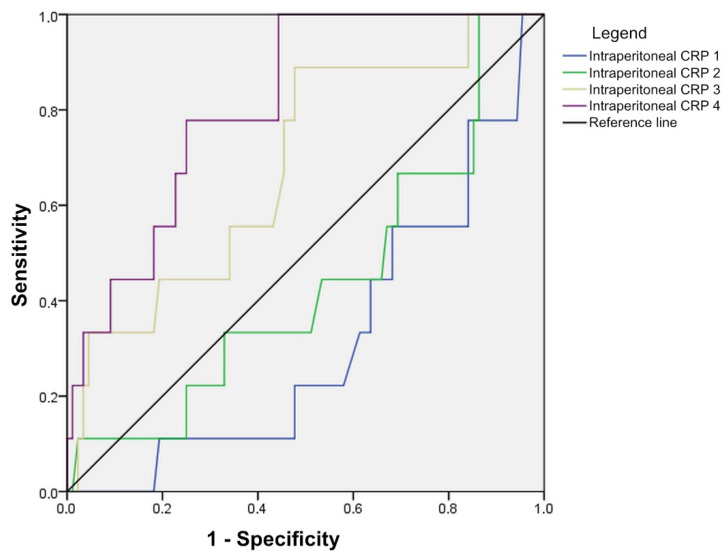
Diagnostic accuracy of intraperitoneal CRP in the detection of anastomotic leak, shown through ROC curves.

**Figure 5 diagnostics-14-02101-f005:**
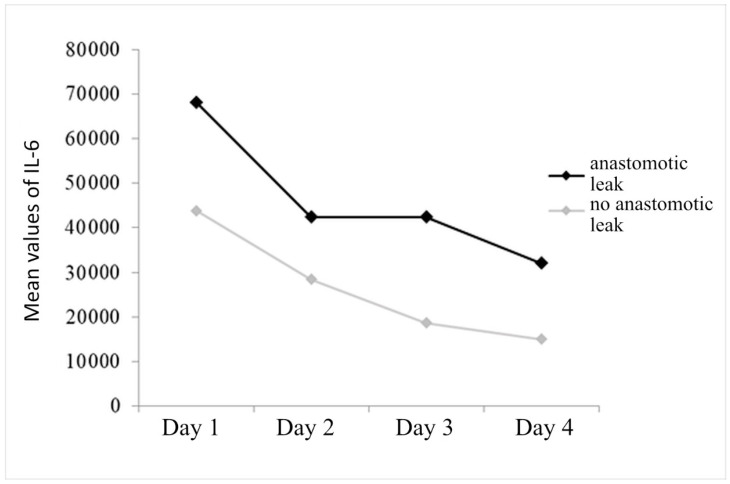
Mean values of interleukin-6 through 4 days of follow-up, with respect to anastomotic leak.

**Figure 6 diagnostics-14-02101-f006:**
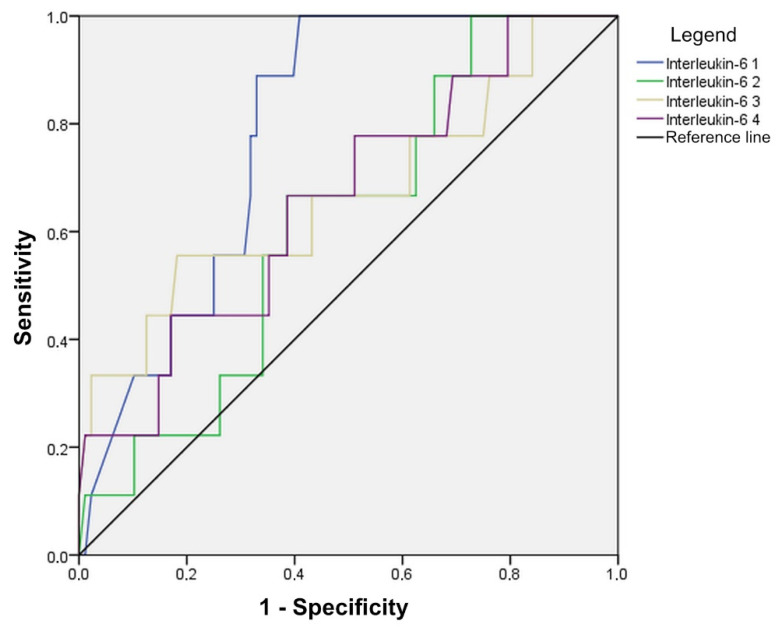
Diagnostic accuracy of IL-6 in anastomotic leak detection, shown through ROC curves.

**Figure 7 diagnostics-14-02101-f007:**
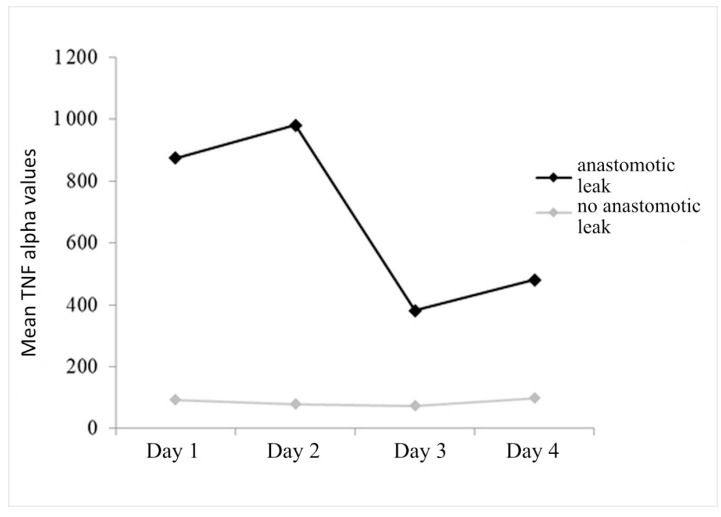
Mean TNF-alpha values over 4 days of follow-up, with respect to dehiscence.

**Figure 8 diagnostics-14-02101-f008:**
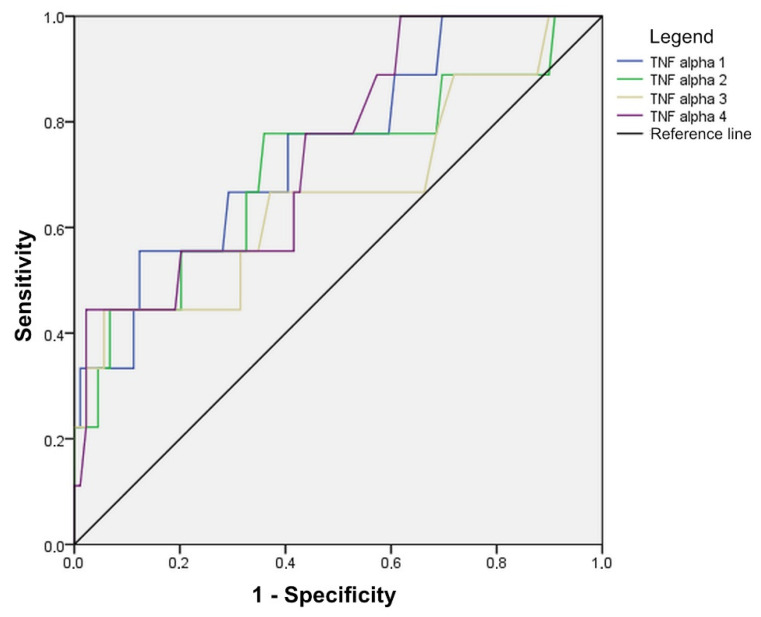
Diagnostic accuracy of TNF-alpha in anastomotic leak detection, shown through ROC curves.

**Table 1 diagnostics-14-02101-t001:** Patients’ characteristics.

		n	(%)
Gender	Male	56	(56.0)
	Female	44	(44.0)
Age (years)	<60	29	(29.0)
	60–79	49	(49.0)
	80+	22	(22.0)
Localisation	Rectum	32	(32.0)
	Left colon	38	(38.0)
	Right colon	30	(30.0)

**Table 2 diagnostics-14-02101-t002:** Cut-off values, sensitivity, specificity, and AUC with associated confidence interval, for serum CRP at 4 post-follow-up days.

	Cut-Off Value	Sensitivity	Specificity	AUC	95% CI	*p*
Serum CRP 1	125.50	60.0	65.2	0.690	0.540–0.839	0.062
Serum CRP 2	162.05	80.0	73.0	0.800	0.684–0.916	0.003
Serum CRP 3	169.90	77.8	79.9	0.853	0.733–0.972	0.001
Serum CRP 4	149.00	77.8	84.3	0.888	0.791–0.984	<0.001

**Table 3 diagnostics-14-02101-t003:** Cut-off values, sensitivity, specificity, and AUC with associated confidence interval, for intraperitoneal CRP at 4 postoperative follow-up days.

	Cut-Off Value	Sensitivity	Specificity	AUC	95% CI	*p*
Peritoneal CRP 1	9.05	40.0	36.0	0.313	0.148–0.479	0.066
Peritoneal CRP 2	31.60	40.0	43.3	0.437	0.238–0.636	0.534
Peritoneal CRP 3	41.15	55.6	57.8	0.687	0.511–0.864	0.065
Peritoneal CRP 4	56.65	77.8	75.6	0.815	0.697–0.933	0.002

**Table 4 diagnostics-14-02101-t004:** Cut-off values, sensitivity, specificity, and AUC with associated confidence interval, for IL-6 at 4 post-follow-up days.

	Cut-Off Value	Sensitivity	Specificity	AUC	95% CI	*p*
Interleukin-6 1	42,150	70.0	67.8	0.782	0.671–0.893	0.005
Interleukin-6 2	26,250	60.0	61.1	0.614	0.445–0.783	0.263
Interleukin-6 3	13,300	55.6	56.7	0.669	0.455–0.883	0.097
Interleukin-6 4	10,800	55.6	62.2	0.662	0.476–0.848	0.111

**Table 5 diagnostics-14-02101-t005:** Cut-off values, sensitivity, specificity, and AUC with associated confidence interval, for TNF-alpha at 4 postoperative follow-up days.

	Cut-Off Value	Sensitivity	Specificity	AUC	95% CI	*p*
TNF-alpha 1	78.00	70.0	66.7	0.749	0.577–0.920	0.014
TNF-alpha 2	58.50	70.0	66.7	0.708	0.504–0.912	0.040
TNF-alpha 3	45.50	66.7	62.2	0.662	0.441–0.881	0.109
TNF-alpha 4	48.50	55.6	57.8	0.746	0.580–0.911	0.015

## Data Availability

The data that have been used are confidential.

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
