# Peer review of "The Value of C-Reactive Protein and Peritoneal Cytokines as Early Predictors of Anastomotic Leak after Colorectal Surgery"

_diagnostics, 2024, doi:10.3390/diagnostics14182101_

Round 1
Reviewer 1 Report
Comments and Suggestions for Authors
I read with interest the manuscript by Mužina et al. titled ‘The value of C-reactive protein and peritoneal cytokines as
early predictors of anastomotic leak after colorectal surgery. The manuscript covers the important topic of anastomotic leak after colorectal surgery with interesting insights on biomarkers to predict leaks.
Please see my comments reported in a point-by-point manner:
- Please include a paragraph on the limitations of the present investigation.
- Please provide a statement of approval from the local ethics committee.
- One limitation of the present investigation is the absence of baseline values for all biomarkers. Baseline values are crucial in studies that involve changes in biomarkers over time, as they provide a reference point and help contextualize the study population.
- Please avoid using the term "unmistakably" in the conclusion, as causality cannot be firmly established given the study's design. Please remain within the scope of hypothesis-driven discussion.
- Maintaining the drain until the fourth post-operative day goes against ERP guidelines and should not be encouraged unless there is demonstrated clinical benefit, which, as of now, has not been supported by high-quality evidence.
- Why did the authors choose to sample IL-6 intra-abdominally instead of through blood samples? Please provide a rationale.
- In addition to the findings, a comment on the surgical stress response and associated biomarkers should be included. Please refer to: Mari G, et al. ERAS Protocol Reduces IL-6 Secretion in Colorectal Laparoscopic Surgery: Results From a Randomized Clinical Trial. Surg Laparosc Endosc Percutan Tech. 2016
Reviewer 2 Report
Comments and Suggestions for Authors
The study design of this article is reasonable, the data analysis is rigorous, and the results have potential clinical application value. With some improvements in detail, its academic impact can be further enhanced.
1. The background section could benefit from including more epidemiological data on anastomotic leaks, as well as the limitations of current diagnostic methods. This would help to highlight the necessity and innovation of this study.
2. Although the methods section is relatively detailed, it would be beneficial to clarify whether the sample size was determined through statistical calculation to ensure sufficient statistical power to detect the occurrence of anastomotic leaks.
3. The discussion could include an analysis of the false positive and false negative results of the biomarkers, as well as specific recommendations on how to use these biomarkers for diagnosis in different clinical scenarios.
4. The study's limitations, such as being a single-center study and having a limited sample size, might affect the generalizability of the results.
5. The conclusion could be strengthened by suggesting future research directions, such as validation in multicenter, large-sample studies, or further investigation of other potential biomarkers to improve the accuracy of predicting anastomotic leaks.
Round 2
Reviewer 2 Report
Comments and Suggestions for Authors
The author has already responded to my related questions and made the necessary modifications.